# Guided-BFNs: Towards Visualizing and Understanding Bayesian Flow Networks in the Context of Trajectory Planning

## Abstract

Bayesian Flow Networks (BFNs) represent an emerging class of generative models that exhibit promising capabilities in modeling continuous, discretized, and discrete data. In this paper, we develop Guided-BFNs to integrate BFNs with conditional guidance and gradient guidance to facilitate the effective application of such models in trajectory planning tasks. Based on our developments, we can better comprehend BFNs by inspecting the generation dynamics of the planning trajectories. Through extensive parameter tuning and rigorous ablation experiments, we systematically delineate the functional roles of various parameters and elucidate the pivotal components within the structure of BFNs. Furthermore, we conduct a comparative analysis of the planning results between diffusion models and BFNs, to discern their similarities and differences. Additionally, we undertake efforts to augment the performance of BFNs, including developing a faster and training-free sampling algorithm for sample generation. Our objectives encompass not only a comprehensive exploration of BFNs' structural insights but also the enhancement of their practical utility.

## 1 Introduction

Generative models (Shocher et al., 2023; Ho et al., 2020; Goodfellow et al., 2014; Kingma & Dhariwal, 2018) have achieved remarkable progress in multimodal generation, with GPTs (OpenAI, 2023; Brown et al., 2020; Radford et al., 2019; 2018) and Stable Diffusion (Rombach et al., 2022; Podell et al., 2023) as the representative examples. The underlying technical foundations include Auto-Regressive Models (Vaswani et al., 2017; Beltagy et al., 2020; Kitaev et al., 2020; Liu et al., 2021; Parmar et al., 2018), Diffusion Models (Ho et al., 2020; Sohl-Dickstein et al., 2015a; Nichol & Dhariwal, 2021; Song et al., 2020a;b; Dhariwal & Nichol, 2021; Ho & Salimans, 2022; Luo, 2022; Ramesh et al., 2022), etc. However, issues exist that the generative modeling of both continuous and discrete data have not been effectively unified.

Bayesian Flow Networks (BFNs) (Graves et al., 2023) are an emerging type of deep generative model for addressing such an issue. They are conceived from the principles of data compression theory (Lelewer & Hirschberg, 1987; Jain, 1981; Welch, 1984). In BFNs, the parameters of a set of independent distributions are modified with Bayesian inference in the light of noisy data samples, then passed as input to a neural network that outputs a second, interdependent distribution. Starting from a simple prior and iteratively updating the two distributions yields a generative procedure similar to the reverse process of diffusion models; however it is conceptually simpler in that no forward process is required. The network inputs for discrete data lie on the probability simplex, and are therefore natively differentiable. The loss function directly optimises data compression and places no restrictions on the network architecture. In the experiments BFNs achieve competitive log-likelihoods for image modelling on dynamically binarized MNIST(LeCun et al., 1998) and CIFAR-10 (Krizhevsky et al., 2009), and outperform all known discrete diffusion models on the text8 (Shannon, 1951; Cover & King, 1978; Zipf, 2013) character-level language modelling task.

However, the previous empirical studies on CIFAR-10 and text8 expose rare insights into the behavior of BFNs and opportunities for further enhancement. We urgently need a good way to better

inspect BFNs and to chase an in-depth understanding, which aids in bringing BFNs to the masses to unleash their maximal potential. This paper takes the first step toward bridging the gap.

We first identify a reasonable way to visualize the internal behavior of BFNs—adapting them for policy modeling and evaluating in long-horizon decision-making scenarios in several reinforcement learning (RL) settings. Given that BFNs employ multi-step sampling in their generation process, akin to diffusion models (DMs), we have chosen to assess them within the context of policy planning tasks for trajectory generation. This decision is made in favor of planning trajectory generation as a testbed for BFNs, primarily because it allows humans to more readily discern and appreciate the properties and nuances of the intermediate planning results. In contrast, generating images involves alterations at the pixel level that are less traceable and exhibit reduced controllability. We choose several RL settings, including the Maze-2D environment (Fu et al., 2020), the block stacking tasks (Garrett et al., 2022), and the D4RL locomotion benchmark (Fu et al., 2020), then launch experiments following the fundamental framework of Diffuser (Janner et al., 2022), a diffusion-based method employed for trajectory planning problem. Under this framework, planning trajectories and sampling from data become essentially equivalent. We will treat "planning" and "sampling" as interchangeable terms in the following paragraphs.

This paper represents the pioneering effort to apply BFNs in RL settings, contributing to the visualization and understanding of BFNs in this context. Our contributions are listed as follows:

First, We introduce a methodology called Guided-BFNs, that integrates additional conditions from the datasets into the model in addressing RL settings using BFNs. In the sampling process, we devise a novel classifier-guided approach to implement gradient-guided and conditional-based sampling with BFNs.

Furthermore, through extensive parameter tuning and comprehensive ablation experiments, we have elucidated the critical components within BFNs' structure and how various parameters function. Utilizing a straightforward and intuitive visualization approach, we visualize the input distribution, output distribution, the sender distribution and the Bayesian update process (Graves et al., 2023), which are the key components of BFNs' sampling process. We also investigated the impact of various factors on BFNs. These factors include the min variance on training loss, planning performance and sample diversity, as well as the interplay between time steps and sample steps during sampling.

Additionally, we conduct a comparative analysis of trajectories generated by both DMs and BFNs within identical RL planning settings. Employing the same network architecture and training parameters allows us to discern the similarities and differences between the two approaches, providing further insights into the functioning of BFNs. Experimental results across various RL settings demonstrate that Guided-BFNs based on the brand new BFNs achieve competitive results compared to extensively optimized DMs.

In the ablation studies, we investigate the effects of the conditional scaling factor and gradient scaling factor on planning performance to demonstrate the effectiveness of the method of Guided-BFNs applied in trajectory planning tasks. These insights significantly contribute to our enhanced understanding of BFNs.

Finally, we tried a interpolation method for training-free faster sampling and compare several interpolation ratios. Our findings on Guided-BFNs in the testbed of RL policy planning settings can be directly transferred to the original testbed focused on image and text data, and we leave that for future research. We hope our findings may shed light on the developing of more effective BFNs' variants and optimization on BFNs' architecture and acceleration on BFNs' sampling process.

## 2 BACKGROUND

In this section, we elucidate the RL policy planning problem setting and provide an overview of BFNs.

### 2.1 PROBLEM SETTING

Consider a system governed by the discrete-time dynamics $s_{t_{p+1}} = f(s_{t_p}, a_{t_p})$ at state $s_{t_p}$ given an action $a_{t_p}$. Trajectory optimization involves determining a sequence of actions $a_{0:T}^*$ that maximizes

(or minimizes) an objective $\mathcal{J}$ factorized over per-timestep rewards (or costs) $r(\boldsymbol{s}_{t_p}, \boldsymbol{a}_{t_p})$:

$$\boldsymbol{a}_{0:T}^* = \arg\min_{\boldsymbol{a}_{0:T}} \mathcal{J}(\boldsymbol{s}_0, \boldsymbol{a}_{0:T}) = \arg\min_{\boldsymbol{a}_{0:T}} \sum_{t_p=0}^{T} r(\boldsymbol{s}_{t_p}, \boldsymbol{a}_{t_p}) \tag{1}$$

where $T$ is the planning horizon and $\boldsymbol{t}_p$ is the planning time step. We use the abbreviation $\boldsymbol{\tau} = (\boldsymbol{s}_0, \boldsymbol{a}_0, \boldsymbol{s}_1, \boldsymbol{a}_1, \ldots, \boldsymbol{s}_T, \boldsymbol{a}_T)$ to refer to a trajectory of interleaved states and actions and $\mathcal{J}(\boldsymbol{\tau})$ to denote the objective value of that trajectory.

## 2.2 BFNs

In our experimental configuration, BFNs undergo an initial learning phase during which they assimilate knowledge pertaining to a comprehensive set of trajectory data, encompassing various state-action pairs. Subsequently, these BFNs harness the acquired knowledge to generate suitable state-action pairs, thereby constituting trajectory data under specific conditions. These conditions may encompass the fulfillment of predefined start and end points within a maze or the stacking of blocks in a predetermined sequence. This subsection will elucidate the fundamental operational principles underlying BFNs.

Assuming each state-action pair in trajectory data adheres to a normal distribution denoted as $\boldsymbol{\theta} \overset{\text{def}}{=} \{\boldsymbol{\mu}, \rho\}$, the input mean $\boldsymbol{\mu}$ (initialized as standard normal $\boldsymbol{\theta}_0 \overset{\text{def}}{=} \{\boldsymbol{0}, 1\}$) is input into BFNs' neural network to obtain the output $\hat{\mathbf{x}}(\boldsymbol{\theta}, t)$, where $t$ represents the time step. Subsequently, both $\boldsymbol{x}$ comprising all state-action pairs of all possible trajectory data in datasets and $\hat{\mathbf{x}}(\boldsymbol{\theta}, t)$ undergo the addition of Gaussian noise following an accuracy schedule determined by $\alpha(t) = -\frac{2 \ln \sigma_1}{\sigma_1^{2t}}$. This results in a sender $\boldsymbol{y}$ and a receiver, where $\sigma_1$ denotes the standard deviation of the input distribution at $t = 1$. The sender $\boldsymbol{y}$ is then employed to update the input parameters $\boldsymbol{\theta}$ through Bayesian inference, given by the equations:

$$\boldsymbol{\mu} \leftarrow \frac{\rho\boldsymbol{\mu} + \alpha\mathbf{y}}{\rho + \alpha}, \rho \leftarrow \rho + \alpha. \tag{2}$$

This iterative process is referred to as Bayesian update. The updated input parameters are subsequently fed into the same neural network, and this process is repeated $n$ times. The Kullback-Leibler (KL) divergence between the sender and receiver is computed for all $n$ iterations, and the sum yields the discrete-time loss. As $n$ approaches infinity, the continuous-time loss is given by:

$$L^{\infty}(\mathbf{x}) = -\ln \sigma_1 \underset{t \sim U(0,1), p_F(\boldsymbol{\theta}|\mathbf{x};t)}{\mathbb{E}} \frac{\|\mathbf{x} - \hat{\mathbf{x}}(\boldsymbol{\theta}, t)\|^2}{\sigma_1^{2t}}. \tag{3}$$

Following training, the neural network of BFNs possesses a learned representation of the original data, encompassing all possible trajectories containing state-action pairs. During the planning process (equivalent to the sample generation process), BFNs generate appropriate trajectories without the need for the original data $\boldsymbol{x}$. The sender $\boldsymbol{y}$ is utilized by introducing Gaussian noise, according to an accuracy schedule, to the output $\hat{\mathbf{x}}(\boldsymbol{\theta}, t)$ for Bayesian updates to the input parameters. This process repeats for $\boldsymbol{S}$ times, with sample steps $s \in \{1, 2, ..., S\}$ corresponding to time step $t \in [0, 1]$.

Detailed information regarding the input distribution $p_I(\mathbf{x} \mid \boldsymbol{\theta})$, output distribution $p_O(\mathbf{x} \mid \boldsymbol{\theta}, t)$, sender distribution $p_S(\cdot \mid \mathbf{x}; \alpha\boldsymbol{I})$, receiver distribution $p_R(\mathbf{y} \mid \boldsymbol{\theta}; t, \alpha)$, Bayesian flow distribution $p_F(\boldsymbol{\theta} \mid \mathbf{x}; t)$, and an overview figure of the training and sampling process of BFNs are provided in the appendix.

In this context, three pivotal time-related variables come into play. The first variable is the sample steps, denoted as $\boldsymbol{s} \in \{1, 2, ..., S\}$, which determines the number of steps taken during the sampling round in BFNs. The second variable is the sample time, denoted as $\boldsymbol{t} \in [0, 1]$, and it is closely associated with the sample steps. The number of sample steps dictates the intervals into which the range $[0, 1]$ is divided during the sampling process in BFNs. The third variable pertains to the planning time steps within the state-action pair, denoted as $\boldsymbol{t_p}$, and spans from 1 to the planning horizon $\boldsymbol{T}$ as defined in eq. (1).

## 3 GUIDED-BFNS

In this section, we establish Guided-BFNs as a novel approach to tackle the planning problem in the upcoming RL experiments. The primary framework, encompassing state-action pairs, temporal locality, trajectory representation, and model architecture, adheres to the structure of Diffuser (Janner et al., 2022). This framework exhibits capabilities such as learning long-horizon planning, temporal compositionality, generation of variable-length plans, and task compositionality.

### 3.1 CONDITIONAL GUIDANCE

Given the parameters of the input distribution $\boldsymbol{\theta}$ and time $t \in [0, 1]$, the output distribution in BFNs is expressed as follows:

$$p_O(\mathbf{x} \mid \boldsymbol{\theta}; t) = \delta(\mathbf{x} - \hat{\mathbf{x}}(\boldsymbol{\theta}, t)), \tag{4}$$

Here, $\delta$ represents the Dirac delta function, and $\hat{\mathbf{x}}(\boldsymbol{\theta}, t)$ corresponds to the output sample obtained after iterating through the neural network for several steps, symbolizing the planning trajectory in this context. For each time step $t \in [0, 1]$ (corresponding to each sample step $\boldsymbol{s} \in \{1, 2, ..., \boldsymbol{S}\}$), BFNs generate a trajectory composed of state-action pairs:

$$\boldsymbol{\tau}_t = \hat{\mathbf{x}}(\boldsymbol{\theta}, t) \tag{5}$$

$$= (\boldsymbol{s}_{t,0}, \boldsymbol{a}_{t,0}, \ldots, \boldsymbol{s}_{t,t_p}, \boldsymbol{a}_{t,t_p}, \ldots, \boldsymbol{s}_{t,T}, \boldsymbol{a}_{t,T}) \tag{6}$$

To incorporate information regarding prior evidence (such as observation history), desired outcomes (like a goal to reach), or general functions to optimize (such as rewards or costs), we introduce the function $h(\boldsymbol{\tau}_t)$ and integrate it into the output sample in eq. (4):

$$p_O(\mathbf{x} \mid \boldsymbol{\theta}; t) \propto p_O(\mathbf{x} \mid \boldsymbol{\theta}; t) h(\boldsymbol{\tau}_t). \tag{7}$$

In certain planning problems, it is more natural to pose them as constraint satisfaction rather than reward maximization. In such settings, the goal is to generate any feasible trajectory that satisfies a set of constraints, such as terminating at a goal location. Representing trajectories in an array, as described by eq. (5), allows translating this setting into an inpainting problem. Here, state and action constraints act similarly to observed pixels in an image (Sohl-Dickstein et al., 2015b). All unobserved locations in the array must be filled in by BFNs in a manner consistent with the observed constraints.

The perturbation function required for this task is a Dirac delta for observed values and constant elsewhere. Specifically, if $\mathbf{c}_t$ is the state constraint at time step $t$, then:

$$h(\boldsymbol{\tau}_t) = \delta_{\mathbf{c}_t}(\boldsymbol{s}_0, \boldsymbol{a}_0, \ldots, \boldsymbol{s}_T, \boldsymbol{a}_T) = \begin{cases} +\infty & \text{if } \mathbf{c}_t = \boldsymbol{s}_t \\ 0 & \text{otherwise} \end{cases}$$

The definition for action constraints is identical. In practice, this may be implemented by sampling from the BFNs' sampling process and replacing the sampled values with conditioning values $\mathbf{c}_t$ after all BFNs sample steps $\boldsymbol{s} \in \{1, 2, ..., \boldsymbol{S}\}$ (or time steps $t \in [0, 1]$). For each planning round, an observed state $\boldsymbol{s}$ is given, and then:

$$\boldsymbol{\tau}_t = \hat{\mathbf{x}}(\boldsymbol{\theta}, t) = (\boldsymbol{s}_{t,0}, \boldsymbol{a}_{t,0}, \ldots, \boldsymbol{s}_{t,t_p}, \boldsymbol{a}_{t,t_p}, \boldsymbol{s}_{t,T}, \boldsymbol{a}_{t,T}) \tag{8}$$

is replaced by

$$\boldsymbol{\tau}_t = \hat{\mathbf{x}}(\boldsymbol{\theta}, t) = (\boldsymbol{s}, \boldsymbol{a}_{t,0}, \ldots, \boldsymbol{s}_{t,t_p}, \boldsymbol{a}_{t,t_p}, \boldsymbol{s}_{t,T}, \boldsymbol{a}_{t,T}) \tag{9}$$

Even in reward maximization problems, conditioning-by-inpainting is necessary because all sampled trajectories should commence from the current state.

### 3.2 GRADIENT GUIDANCE

Given the original data $\boldsymbol{x}$ and accuracy $\alpha$, the sender distribution is defined as follows:

$$p_S(\mathbf{y} \mid \mathbf{x}; \alpha) = \mathcal{N}\left(\mathbf{y} \mid \mathbf{x}, \alpha^{-1}\boldsymbol{I}\right). \tag{10}$$

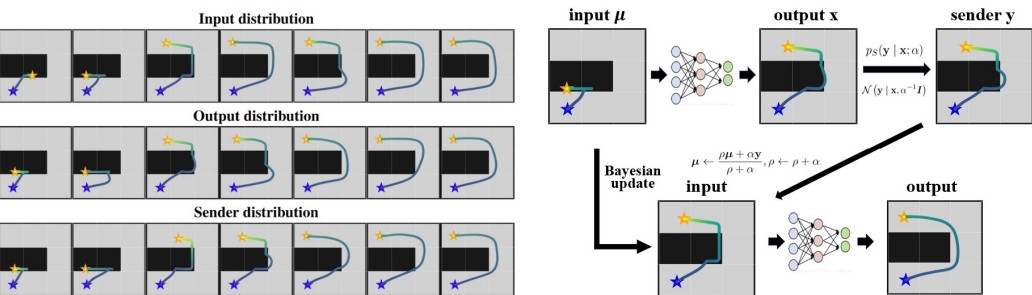

Figure 1: This figure illustrates the functioning of several key **components** in BFNs through visualization on Maze-2D. **Left:** Visualization of the input distribution $p_I$, the output distribution $p_O$, and the sender distribution $p_S$ as sample steps $s$ increases. Definitions of these distributions in BFNs are described in section 2.2, with additional mathematical details available in the appendix. All three distributions gradually update towards the correct trajectory, utilizing different functions. **Right:** Visualization of how these key components in BFNs work in a single sample step. The neural network addresses interrelated variables in the data by observing that output trajectories tend to differ significantly from the input ones, indicating large transitions to the correct trajectory. The Bayesian update process deals with independent variables under statistical theory assumptions, revising the trajectory in a more moderate way towards the correct outcome.

To address RL problems using BFNs, the introduction of a reward concept is imperative. We employ the control-as-inference graphical model (Levine, 2018) for this purpose. Let $\mathcal{O}_{t_p}$ be a binary random variable denoting the optimality of planning timestep $t_p$ of a trajectory, with $p(\mathcal{O}_{t_p} = 1) = \exp(r(\boldsymbol{s}_{t_p}, \boldsymbol{a}_{t_p}))$. We can sample from the set of optimal trajectories by setting $h(\boldsymbol{\tau}_t) = p(\mathcal{O}_{1:T} \mid \boldsymbol{\tau}_t)$ in eq. (7):

$$p_O(\mathbf{x} \mid \boldsymbol{\theta}; t) = p(\boldsymbol{\tau}_t \mid \mathcal{O}_{1:T} = 1) \tag{11}$$
$$\propto p_O(\mathbf{x} \mid \boldsymbol{\theta}; t) p(\mathcal{O}_{1:T} = 1 \mid \boldsymbol{\tau}_t). \tag{12}$$

The initial step involves training a BFNs' model on the states and actions encompassed in all available trajectory data. Subsequently, a separate model denoted as $\mathcal{J}_\phi$ is trained to predict the cumulative rewards of the trajectory $\boldsymbol{\tau}_t = \hat{\mathbf{x}}(\boldsymbol{\theta}, t)$ sampled from $p_O(\mathbf{x} \mid \boldsymbol{\theta}; t)$. The gradients of $\mathcal{J}_\phi$ play a pivotal role in guiding the trajectory sampling procedure by modifying the input means $\boldsymbol{\mu}$, output sample $\hat{\mathbf{x}}(\boldsymbol{\theta}, t)$, and the sender distribution $p_S(\mathbf{y} \mid \mathbf{x}; \alpha)$ of the sampling process. This modification is carried out based on the min variance $\boldsymbol{\sigma_1}$, sampling time $t \in [0, 1]$, and accuracy $\alpha$ according to the following equations:

$$\boldsymbol{\mu}_{\text{new}} = \boldsymbol{\mu}_{\text{last}} + \sigma_1^t \cdot g \tag{13}$$
$$\hat{\mathbf{x}}(\boldsymbol{\theta}, t)_{\text{new}} = \hat{\mathbf{x}}(\boldsymbol{\theta}, t)_{\text{last}} + \sigma_1^t \cdot g \tag{14}$$
$$p_S(\mathbf{y} \mid \hat{\mathbf{x}}(\boldsymbol{\theta}, t); \alpha) = \mathcal{N}\left(\mathbf{y} \mid \hat{\mathbf{x}}(\boldsymbol{\theta}, t) + \sigma_1^t \cdot g, \alpha^{-1} \boldsymbol{I}\right) \tag{15}$$

where

$$g = \nabla_{\boldsymbol{\tau}_t} \log p(\mathcal{O}_{1:T} \mid \boldsymbol{\tau}_t)|_{\boldsymbol{\tau}_t = \hat{\mathbf{x}}(\boldsymbol{\theta}, t)} \tag{16}$$
$$= \sum_{t_p=0}^{T} \nabla_{\boldsymbol{s}_{t_p}, \boldsymbol{a}_{t_p}} r(\boldsymbol{s}_{t_p}, \boldsymbol{a}_{t_p})|_{(\boldsymbol{s}_{t_p}, \boldsymbol{a}_{t_p}) = \hat{\mathbf{x}}(\boldsymbol{\theta}, t)} \tag{17}$$
$$= \nabla \mathcal{J}_\phi(\hat{\mathbf{x}}(\boldsymbol{\theta}, t)). \tag{18}$$

The rationale for incorporating the gradient into $\boldsymbol{\mu}$, $\hat{\mathbf{x}}(\boldsymbol{\theta}, t)$, and the sender distribution $p_S(\mathbf{y} \mid \hat{\mathbf{x}}(\boldsymbol{\theta}, t); \alpha)$ is twofold. Firstly, $\hat{\mathbf{x}}(\boldsymbol{\theta}, t)$ inherently contains information about the planning trajectory and thus requires guidance from the reward function. Secondly, in the Bayesian update function in BFNs (Graves et al., 2023), represented by eq. (2), both $\boldsymbol{\mu}$ and $\boldsymbol{y}$ in the numerator influence the update of the input mean transmitted to the neural network. This, in turn, affects future parameter updates and the data processed by the model. Therefore, they must also be guided by the reward function $\mathcal{J}_\phi$.

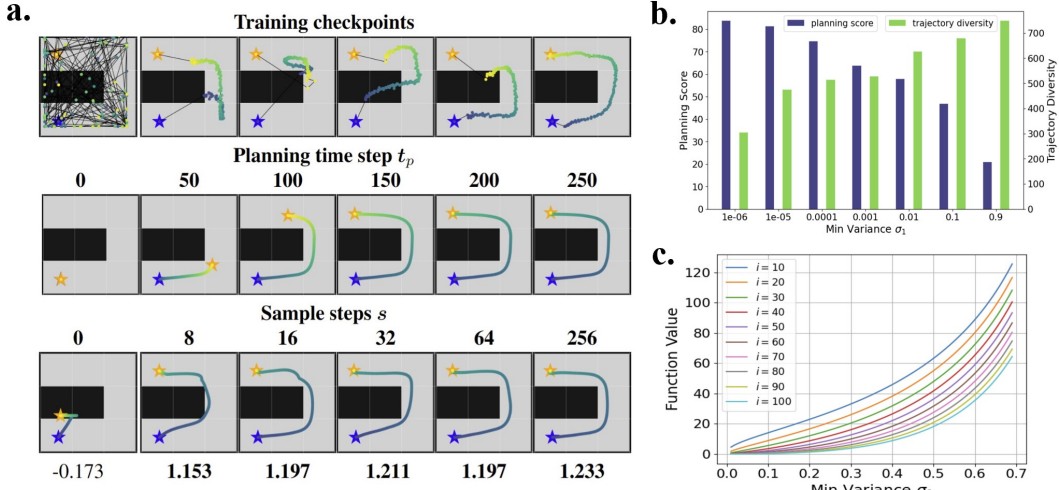

Figure 2: This figure provides an overview of the influence of key **parameters** in BFNs. **a.** By selecting different checkpoints during training, the figure visualizes how the correct trajectory is learned and updated by BFNs. Planning performance increases as planning time step $t_p$ and sample steps $s$ increase. The last line illustrates the planning score according to sample steps. **b.** As min variance $\sigma_1$ increases, the planning score decreases, but the generated trajectories' diversity (measured by the variance of 100 different trajectories) increases. This finding enlightens similar results in the original testbed of image and text data in BFNs. **c.** The variance of the sender distribution increases for a fixed sample steps $S = 100$ as $\sigma_1$ increases for all discrete sample steps $s = n \in \{1, 2, ..., S\}$. This explains the sampling diversity result in (**b**) and the decreasing speed of trajectory shape variation in fig. 1 **Left**.

It is noteworthy that, in addition to the gradient, we introduce a posterior variance $\sigma_1^t$. Theoretically, incorporating a posterior variance $\sigma_1^t$ imparts varying degrees of information based on different variances and times, resulting in more effective guidance similar to classifier-guidance (Dhariwal & Nichol, 2021) in DMs. Empirically, our experiments demonstrate a substantial improvement in planning performance with the inclusion of posterior variance $\sigma_1^t$.

Finally, the first action of a sampled trajectory $\tau_t \sim p(\tau_t \mid \mathcal{O}_{1:T} = 1)$ may be executed in the environment, after which the planning procedure recommences in a standard receding-horizon control loop.

### 3.3 TRAINING AND SAMPLING

The training step of Guided-BFNs directly optimizes the continuous time loss in eq. (3). Conditional guidance is combined in the training process. The model architecture, the reward model $\mathcal{J}_\phi$ and other training details are listed in appendix. The rationale behind training the reward function lies in utilizing the gradient of the output sample to guide trajectory generation, representing a novel approach in classifier-guided BFNs.

The sampling step of Guided-BFNs combines both the conditional guidance and the gradient guidance from the reward function $\mathcal{J}_\phi$. Pseudocode for the training method and guided planning method is given in algorithms 1 to 3 in appendix.

## 4 EXPERIMENTS

In this section, we perform a series of experiments aimed at visualizing and comprehending the behavior of BFNs with Guided-BFNs in the RL policy planning setting. Our investigations encompass several key aspects. Initially, key components of BFNs' architecture, including the input distribution, the output distribution and the sender distribution are visualized under the Maze-2D setting to investigate how the neural network and Bayesian update function work. Subsequently, we delve into

the influence of the min variance $\sigma_1$, the planning time steps $t_p$ and sample steps $s$ on training loss, planning score and samples diversity. Following this, we conduct a comparative analysis of trajectories generated by both DMs and BFNs to unveil their similarities and differences. We then proceed with ablation studies, demonstrating the successful operation of the novel Guided-BFNs method under both conditional and gradient guidance. Finally, we explore interpolation studies, aiming to develop a training-free and faster sampling method for BFNs.

To quantitatively assess planning effectiveness, we introduce a metric called **score**. A higher score indicates superior model performance, representing the normalized cumulative reward eq. (1) of the agent up to the current time step in the episode. In the reinforcement learning environment of Mujoco, the score can be directly calculated for the trajectory. Taking the Maze-2D environment as an example, the score calculation includes whether the target is reached, the path length, the number of time steps to reach the target, and the number of collisions. A larger score indicates a better trajectory.

**Datasets** We utilize the Maze-2D (Fu et al., 2020), Kuka block stacking (Garrett et al., 2022), and D4RL locomotion datasets (Brockman et al., 2016; Fu et al., 2020) in the RL setting with Guided-BFNs to visualize and comprehend the behavior of BFNs. The introduction of these datasets and details of $h(\tau)$ and reward settings of each dataset are listed in appendix.

### 4.1 KEY COMPONENTS IN BFNs

In this section, we provide visualizations of the input distribution $p_I$, the output distribution $p_O$, and the sender distribution $p_S$ (as discussed in section 2.2), presented in fig. 1. Additional mathematical details are available in the appendix. These visualizations aim to elucidate the workings of key components within BFNs.

The parameters of an input distribution (Gaussian, with both mean and variance, but only the mean $\mu$ is considered) are inputted and processed by a pretrained neural network optimized using the continuous time loss in eq. (3). This process yields the parameters of an output distribution (Delta distribution). Noise is subsequently added to the output, resulting in a sender $y$, according to the accuracy schedule outlined in section 2.2 and the appendix. Following this, the input and sender are utilized to update the initial input distribution through Bayesian inference. The updated input parameters are then fed into the same neural network, producing the final output, which is the planning trajectory in our setting.

**Neural Network:** The neural network addresses interrelated variables in data by leveraging its capacity to amalgamate such variables and learn implicit representations. This is a fundamental aspect observed in deep learning, where the output trajectories exhibit significant deviations from the input ones, representing large transitions toward the correct trajectory.

**Bayesian Update:** The Bayesian update process deals with independent variables in accordance with statistical theory. Trajectories undergo more moderate revisions towards the correct outcome through this process. The combination of these components in BFNs harnesses the advantages of both deep learning and Bayesian inference.

### 4.2 KEY PARAMETERS IN BFNs

In this section, we explore the impact of several key parameters in BFNs, as shown in fig. 2. These parameters include the min variance $\sigma_1$, planning time step $t_p$, and sample steps $s$ through visualization.

**Min Variance $\sigma_1$** The hyperparameter $\sigma_1$ is of paramount importance in BFNs, influencing the accuracy schedule, which determines the manner and rate of introducing noise to the original data during both training and inference. Specifically, $\sigma_1$ represents the standard deviation of the input distribution at $t = 1$ in eq. (3). Although the original BFNs paper (Graves et al., 2023) provides an approach for deriving the accuracy schedule, the selection process for $\sigma_1$ is not discussed or experimented within the context of BFNs.

In our observations, we note that the converged training loss tends to decrease as $\sigma_1$ increases, as shown in table 8 in the appendix. Simultaneously, the fluctuation of loss during training shows an inverse relationship with $\sigma_1$; it increases as $\sigma_1$ decreases.

After training on different values of $\boldsymbol{\sigma_1}$, we evaluate planning performance with these pretrained Guided-BFNs with fixed sample steps $\boldsymbol{S} = 64$ on 100 trajectories generated by each $\boldsymbol{\sigma_1}$. We also investigate the samples' diversity, measured by the variance of 100 trajectories' state-action pairs data (fig. 2 **(b)**). We find that as $\boldsymbol{\sigma_1}$ increases, the planning performance decreases, while the samples' diversity increases. There is a trade-off between sampling quality (correctness of trajectory) and sampling diversity (different shapes of trajectory) determined by $\boldsymbol{\sigma_1}$ in BFNs. This finding is valuable for designing testbeds for BFNs with image and text data, where the influence of $\boldsymbol{\sigma_1}$ on sample quality and diversity is opposite and requires careful consideration.

The variance of the sender distribution in eq. (10), determined by the accuracy schedule during BFNs' sampling, is defined in algorithm 3 in the appendix:

$$\alpha = \sigma_1^{-2i/n} \left( 1 - \sigma_1^{2/n} \right) \tag{19}$$

where $n = \boldsymbol{S}$ is the total sample steps, and $i = \boldsymbol{s} \in \{1, 2, ..., n\}$. We visualize this variance with different $\boldsymbol{\sigma_1}$ and $i$ in fig. 2 **(c)**. As $\boldsymbol{\sigma_1}$ increases, the variance of the sender distribution rises, which is used for Bayesian update (fig. 1 **Right**). Consequently, the diversity of samples increases, as confirmed in fig. 2 **(b)**. Additionally, as sample steps increase, the variance decreases with a fixed $\boldsymbol{\sigma_1}$, leading to less uncertainty in Bayesian update. This results in less change in input, output, and sender distribution as sample steps increase, as intuitively confirmed in fig. 1 **Left**.

**Planning Time Steps $\boldsymbol{t_p}$ & Sample Steps $\boldsymbol{s}$** In the sampling process of Guided-BFNs, two crucial parameters, planning time steps $\boldsymbol{t_p}$ in the RL setting and sample steps $\boldsymbol{s}$ in BFNs during the planning round, play a vital role. The trajectory's gradual approach towards the goal is illustrated in fig. 2 **(a)**. Additionally, we observe an improvement in planning performance as both $\boldsymbol{t_p}$ and $\boldsymbol{s}$ increase. These findings provide valuable insights for parameter selection, contributing to a deeper understanding of BFNs.

## 4.3 BFNs vs DMs

Table 1: Planning performance under the same setting between Diffuser Janner et al. (2022) based on DMs and Guided-BFNs based on BFNs Graves et al. (2023).

| Dataset / Method / Score | Diffuser | **Guided-BFNs** |
| --- | --- | --- |
| Maze2D-Umaze | 113.9 | **121.4** |
| Maze2D-Medium | 121.5 | **132.5** |
| Maze2D-Large | 123.0 | **136.5** |
| Kuka-Unconditional | 58.7 | 57.7 |
| Kuka-Conditional | 45.6 | **50.2** |
| Kuka-Rearrangement | 58.9 | **63.4** |
| halfcheetah-medium-expert-v2 | 88.9 | **98.5** |
| halfcheetah-medium-replay-v2 | 37.7 | **50.7** |
| halfcheetah-medium-v2 | 42.8 | **64.1** |
| hopper-medium-expert-v2 | 103.3 | **110.7** |
| hopper-medium-replay-v2 | 93.6 | **100.0** |
| hopper-medium-v2 | 74.3 | **85.4** |
| walker2d-medium-expert-v2 | 106.9 | **118.6** |
| walker2d-medium-replay-v2 | 70.6 | **74.1** |
| walker2d-medium-v2 | 79.6 | **82.6** |

Upon scrutinizing the effects of components and parameters in BFNs, we extend our investigation by conducting comparative studies between DMs and BFNs applied to the same task. This analysis aims to elucidate their respective trajectories, seeking to identify similarities and differences, thereby enhancing our understanding of BFNs.

**Unlimited Sampling Steps** Our investigation reveals a distinct advantage of BFNs over DMs. BFNs exhibit flexibility by enabling an unrestricted number of sample steps during planning, facilitated by the utilization of a continuous time loss function (3). This feature eliminates the need for a fixed step during training, allowing for the enhancement of planning performance with an increase in sample steps $\boldsymbol{S}$. In contrast, DMs impose limitations, permitting only a finite number of sampling steps equal to those fixed during the training process.

**Different Trajectories** Additionally, we conduct a comparative analysis of the trajectories learned by both DMs and BFNs from the provided data. To gain insights into how these models process

disorganized data, we visualize the results in fig. 3 **Left**, shedding light on the distinct approaches employed by DMs and BFNs in handling complex and unstructured datasets.

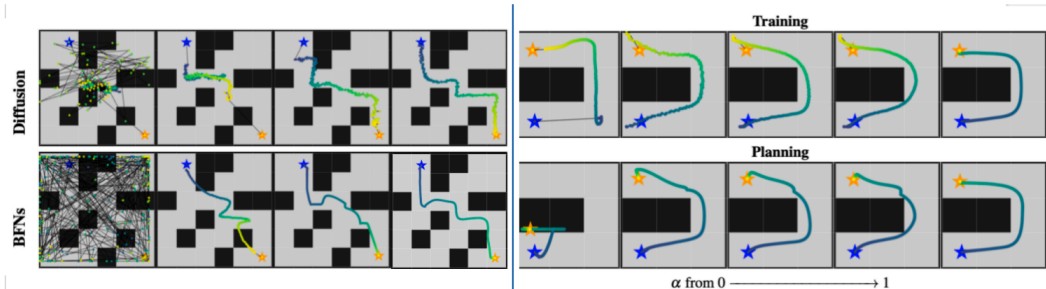

Figure 3: **Left:** The initial data is more noisy for BFNs than DMs. However BFNs can generate proper trajectory more faster than DMs, which indicates a more rapid learning speed for BFNs. The trajectory generated by BFNs is more smooth that DMs. BFNs learn by Bayesian update while DMs learn by denoising data. **Right:** As the conditional ratio $\alpha$ incrementally approaches 1, indicative of an increased reliance on conditioning to guide the trajectories towards the start and end points, a noticeable trend emerges.

**Competitive Results** In a conclusive evaluation, we assess the planning performance under identical training and sampling configurations using both Diffuser (Janner et al., 2022) and Guided-BFNs in table 1. The outcomes indicate that Guided-BFNs not only achieve competitive performance but, in most instances, surpass the performance achieved by DMs. This attests to the efficacy of Guided-BFNs in delivering robust planning outcomes, establishing them as a compelling alternative to DMs in the context of the evaluated tasks.

### 4.4 ABLATION STUDIES

To assess the effectiveness of the novel method Guided-BFNs, incorporating both conditional and gradient guidance (analogous to classifier-guidance (Dhariwal & Nichol, 2021) in DMs), ablation studies are conducted. For conditional guidance, we introduce a scaling factor $\alpha \in [0, 1]$ in front of the conditional guidance term in eq. (7), transforming it into:

$$p_O(\mathbf{x} \mid \boldsymbol{\theta}; t) = \alpha p_O(\mathbf{x} \mid \boldsymbol{\theta}; t) h(\boldsymbol{\tau}).$$

In practice, we directly multiply the state-action pair trajectory array eq. (9) by $\alpha$. We examine how the trajectory evolves as $\alpha$ varies, specifically on the Maze2d-umaze dataset, as depicted in fig. 3.

Simultaneously, for the gradient guidance in Guided-BFNs, we introduce a scalar $\alpha'$ in eq. (13), resulting in:

$$p_S(\mathbf{y} \mid \hat{\mathbf{x}}(\boldsymbol{\theta}, t); \alpha) \approx \mathcal{N}\left(\mathbf{y} \mid \hat{\mathbf{x}}(\boldsymbol{\theta}, t) + \boldsymbol{\sigma}_1^t * g * \alpha', \alpha^{-1}\boldsymbol{I}\right).$$

We investigate the impact of the gradient scalar $\alpha'$ on planning score and conduct experiments on Kuka-conditioning and Kuka-rearrangement datasets, as illustrated in fig. 4. This finding underscores the pivotal role of gradient guidance in achieving superior planning outcomes in the evaluated scenarios.

### 4.5 INTERPOLATION SAMPLING ACCELERATION

Developing training-free, faster sampling algorithms for generative AI models is of paramount importance. In our pursuit of this objective, we experimented with an interpolation method applied to BFNs on the Kuka-unconditioning dataset. The aim was to accelerate the sampling process and potentially provide insights for future research in the domain of efficient BFNs sampling methods.

In essence, we kept the last sample step $\boldsymbol{S}$ unchanged and evenly distributed sampling points at equal intervals among the remaining $\boldsymbol{S} - \boldsymbol{1}$ points. The details of the interpolation sampling acceleration algorithm can be found in the appendix. The remarkable outcome signifies an acceleration in sampling efficiency by half (fig. 4), showcasing the potential of interpolation as a promising strategy for expediting the sampling process without compromising planning performance.

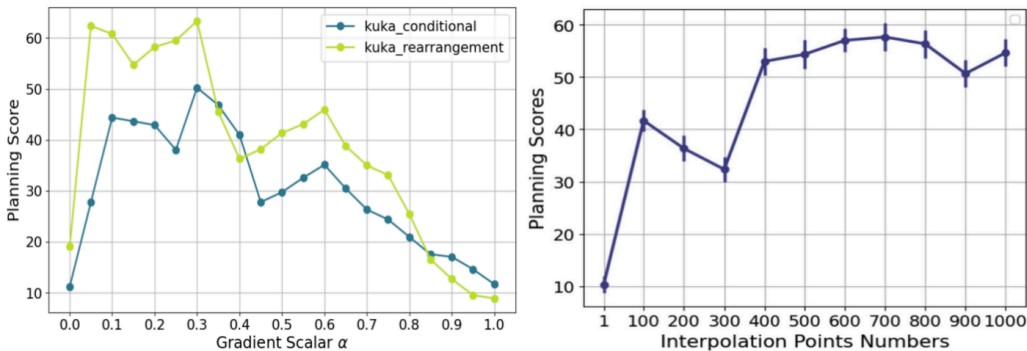

Figure 4: **Left:** The depicted figure illustrates the variation in planning scores corresponding to different gradient scalar values $\alpha'$ within the Kuka settings. Our observations reveal that a non-zero scalar $\alpha'$ significantly enhances planning performance compared to the case where $\alpha'$ is set to 0, thereby validating the effectiveness of gradient guidance. Notably, the optimal gradient scalar under this specific setting is determined to be 0.3. **Right:** The depicted figure illustrates the impact of the interpolation method on the planning score, as observed in experiments conducted on the Kuka-unconditional dataset. The experiments were conducted using a pretrained Guided-BFNs with a fixed sample step of $S = 1000$ on the average of 1000 trajectories each interpolation points numbers. The x-axis represents the number of sample steps preserved after interpolation. By employing interpolation, we can preserve approximately half of the original sample steps while maintaining nearly the same planning score as observed without interpolation.

## 5 RELATED WORK

Deep generative modeling has significantly advanced model-based reinforcement learning. Recent studies explore dynamic models with neural ODEs (Du et al., 2020), vector quantized autoencoders (Hafner et al., 2020), and Transformers (Chen et al., 2022). This reflects a shift in the research focus and methods. Various works (Tamar et al., 2017; Farahmand et al., 2017; Rybkin et al., 2021) have investigated bridging the gap between model learning and planning. Notably, (Janner et al., 2022) introduced Diffuser, a diffusion (Ho et al., 2020; Sohl-Dickstein et al., 2015a; Nichol & Dhariwal, 2021)-based model that concurrently generates trajectory timesteps, conditioned with auxiliary functions. This paper pioneers in visualizing and comprehending Bayesian Flow Networks (BFNs) (Graves et al., 2023). Their adaptability to continuous, discretized, and discrete data, with minimal training adjustments, stands in contrast to discretized diffusion models (Austin et al., 2021), which necessitate defined transition matrices.

## 6 CONCLUSION

In summary, this paper introduces Guided-BFNs, an innovative extension of BFNs, showcasing their enhanced applicability in various RL scenarios through the integration of additional conditions and gradient guidance. Through systematic parameter tuning and rigorous experiments, we uncover crucial components and elucidate the functional roles of parameters, providing valuable insights for practitioners. A comparative analysis highlights difference between BFNs and DMs. Our contribution includes a faster, training-free sampling algorithm, improving efficiency for real-time applications. By presenting our findings, we aim to inspire further research in the realms of BFNs' explanation and optimization, fostering advancements in these areas in the future.

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

## A  COMPUTATIONAL COMPLEXITY

We have supplemented relevant experiments on the computational complexity of BFNs. Taking the maze2d-umaze dataset as an example, on a single NVIDIA 4090 GPU:

| | Time (1000 steps) | GPU Memory | FLOPs (1000 steps) | Parameters |
|---|---|---|---|---|
| Training | | | | |
| Diffuser | 15.7893s | 1152M | 4.189G | 3.675M |
| Guided-BFNs | 16.9726s | 1152M | 4.189G | 3.675M |
| Sampling | | | | |
| Diffuser | 0.5228s | 546M | 8.378G | 3.675M |
| Guided-BFNs | 0.6141s | 548M | 59.026G | 2.652M |

Table 2: Comparison of Train and Sample Time, GPU Memory, FLOPs, and Parameters for Diffuser and Guided-BFNs

From the above data, we can see that in the trajectory planning scenario, in terms of computational complexity, BFNs have almost the same indicators as diffusion models during training. However, in the sampling phase, the FLOPs of BFNs are significantly higher than diffusion models. This is because in addition to the step of updating the input and output parameters by the neural network, BFNs have the Bayesian update process (Equation 2) that diffusion models lack, requiring three additional vector multiplications and vector additions per sample step.

## B  DETAILS OF GUIDED-BFNS

### B.1  MATHEMATICAL DETAILS OF BFNS

In our experiments, $x$ is normalised to lie in $[-1, 1]^D$ to ensure that the network inputs remain in a reasonable range. The input distribution for continuous data is a diagonal normal:

$$\boldsymbol{\theta} \stackrel{\text{def}}{=} \{\boldsymbol{\mu}, \rho\} \tag{20}$$

$$p_I(\mathbf{x} \mid \boldsymbol{\theta}) \stackrel{\text{def}}{=} \mathcal{N}\left(\mathbf{x} \mid \boldsymbol{\mu}, \rho^{-1}\boldsymbol{I}\right), \tag{21}$$

where $\boldsymbol{I}$ is the $D \times D$ identity matrix. We define the prior parameters as

$$\boldsymbol{\theta}_0 \stackrel{\text{def}}{=} \{\mathbf{0}, 1\} \tag{22}$$

where $\mathbf{0}$ is the length $D$ vectors of zeros. Hence the input prior is a standard multivariate normal:

$$p_I\left(\mathbf{x} \mid \boldsymbol{\theta}_0\right) = \mathcal{N}(\mathbf{x} \mid \mathbf{0}, \boldsymbol{I}). \tag{23}$$

Bayesian update function $h\left(\boldsymbol{\theta}_{i-1}, \mathbf{y}, \alpha\right)$ obtains the parameters $\boldsymbol{\theta}_{i-1} = \{\boldsymbol{\mu}_{i-1}, \rho_{i-1}\}$ and sender sample $\mathbf{y}$ drawn from $p_S(\cdot \mid \mathbf{x}; \alpha\boldsymbol{I}) = \mathcal{N}\left(\mathbf{x}, \alpha^{-1}\boldsymbol{I}\right)$ :

$$h\left(\{\boldsymbol{\mu}_{i-1}, \rho_{i-1}\}, \mathbf{y}, \alpha\right) = \{\boldsymbol{\mu}_i, \rho_i\}, \tag{24}$$

with

$$\rho_i = \rho_{i-1} + \alpha, \tag{25}$$

$$\boldsymbol{\mu}_i = \frac{\boldsymbol{\mu}_{i-1}\rho_{i-1} + \mathbf{y}\alpha}{\rho_i}. \tag{26}$$

Bayesian update distribution has form

$$p_U\left(\boldsymbol{\theta}_i \mid \boldsymbol{\theta}_{i-1}, \mathbf{x}; \alpha\right) = \mathcal{N}\left(\boldsymbol{\mu}_i \mid \frac{\alpha\mathbf{x} + \boldsymbol{\mu}_{i-1}\rho_{i-1}}{\rho_i}, \frac{\alpha}{\rho_i^2}\boldsymbol{I}\right) \tag{27}$$

Accuracy schedule can be described as

$$\sigma_1^2 = (1 + \beta(1))^{-1}.(1 + \beta(t))^{-1} = \sigma_1^{2t} \tag{28}$$

$$\implies \beta(t) = \sigma_1^{-2t} - 1 \tag{29}$$

$$\implies \alpha(t) = \frac{d\left(\sigma_1^{-2t} - 1\right)}{dt} \tag{30}$$

$$= -\frac{2\ln\sigma_1}{\sigma_1^{2t}} \tag{31}$$

Bayesian flow distribution can be described as

$$p_F(\boldsymbol{\theta} \mid \mathbf{x}; t) = p_U\left(\boldsymbol{\theta} \mid \boldsymbol{\theta}_0, \mathbf{x}, \beta(t)\right) \tag{32}$$

Therefore, setting $\boldsymbol{\theta}_{i-1} = \boldsymbol{\theta}_0 = \{\mathbf{0}, 1\}$ and $\alpha = \beta(t)$, and that $\rho = 1 + \beta(t)$ ,

$$p_F(\boldsymbol{\theta} \mid \mathbf{x}; t) = \mathcal{N}\left(\boldsymbol{\mu} \mid \frac{\beta(t)}{1 + \beta(t)}\mathbf{x}, \frac{\beta(t)}{(1 + \beta(t))^2}\boldsymbol{I}\right) \tag{33}$$

$$= \mathcal{N}(\boldsymbol{\mu} \mid \gamma(t)\mathbf{x}, \gamma(t)(1 - \gamma(t))\boldsymbol{I}), \tag{34}$$

where

$$\gamma(t) \overset{\text{def}}{=} \frac{\beta(t)}{1 + \beta(t)} \tag{35}$$

$$= \frac{\sigma_1^{-2t} - 1}{\sigma_1^{-2t}} \tag{36}$$

$$= 1 - \sigma_1^{2t}. \tag{37}$$

Output distribution can be discribed as Following standard practice for diffusion models, the output distribution is defined by reparameterising a prediction of the Gaussian noise vector $\boldsymbol{\epsilon} \sim \mathcal{N}(\mathbf{0}, \boldsymbol{I})$ used to generate the mean $\boldsymbol{\mu}$ passed as input to the network.

$$\boldsymbol{\mu} \sim \mathcal{N}(\gamma(t)\mathbf{x}, \gamma(t)(1 - \gamma(t))\boldsymbol{I}) \tag{38}$$

and hence

$$\boldsymbol{\mu} = \gamma(t)\mathbf{x} + \sqrt{\gamma(t)(1 - \gamma(t))}\boldsymbol{\epsilon} \tag{39}$$

$$\implies \mathbf{x} = \frac{\boldsymbol{\mu}}{\gamma(t)} - \sqrt{\frac{1 - \gamma(t)}{\gamma(t)}}\boldsymbol{\epsilon}. \tag{40}$$

The network outputs an estimate $\hat{\boldsymbol{\epsilon}}(\boldsymbol{\theta}, t)$ of $\boldsymbol{\epsilon}$ and this is transformed into an estimate $\hat{\mathbf{x}}(\boldsymbol{\theta}, t)$ of $\mathbf{x}$ by

$$\hat{\mathbf{x}}(\boldsymbol{\theta}, t) = \frac{\boldsymbol{\mu}}{\gamma(t)} - \sqrt{\frac{1 - \gamma(t)}{\gamma(t)}}\hat{\boldsymbol{\epsilon}}(\boldsymbol{\theta}, t). \tag{41}$$

Given $\hat{\boldsymbol{x}}(\boldsymbol{\theta}, t)$ the output distribution is

$$p_O(\mathbf{x} \mid \boldsymbol{\theta}; t) = \delta(\mathbf{x} - \hat{\mathbf{x}}(\boldsymbol{\theta}, t)), \tag{42}$$

Sender distribution can be discribed as The sender space $\mathcal{Y} = \mathcal{X} = \mathbb{R}$ for continuous data, and the sender distribution is normal with precision $\alpha$ :

$$p_S(\mathbf{y} \mid \mathbf{x}; \alpha) = \mathcal{N}\left(\mathbf{y} \mid \mathbf{x}, \alpha^{-1}\boldsymbol{I}\right). \tag{43}$$

Receiver distribution can be described as

$$p_R(\mathbf{y} \mid \boldsymbol{\theta}; t, \alpha) = \underset{\delta(\mathbf{x}' - \hat{\mathbf{x}}(\boldsymbol{\theta}, t))}{\mathbb{E}} \mathcal{N}\left(\mathbf{y} \mid \mathbf{x}', \alpha^{-1}\boldsymbol{I}\right) \tag{44}$$

$$= \mathcal{N}\left(\mathbf{y} \mid \hat{\mathbf{x}}(\boldsymbol{\theta}, t), \alpha^{-1}\boldsymbol{I}\right). \tag{45}$$

Continuous time loss can be described as

$$L^\infty(\mathbf{x}) = -\ln\sigma_1 \underset{t\sim U(0,1), p_F(\boldsymbol{\theta}\mid\mathbf{x};t)}{\mathbb{E}} \frac{\|\mathbf{x} - \hat{\mathbf{x}}(\boldsymbol{\theta}, t)\|^2}{\sigma_1^{2t}} \tag{46}$$

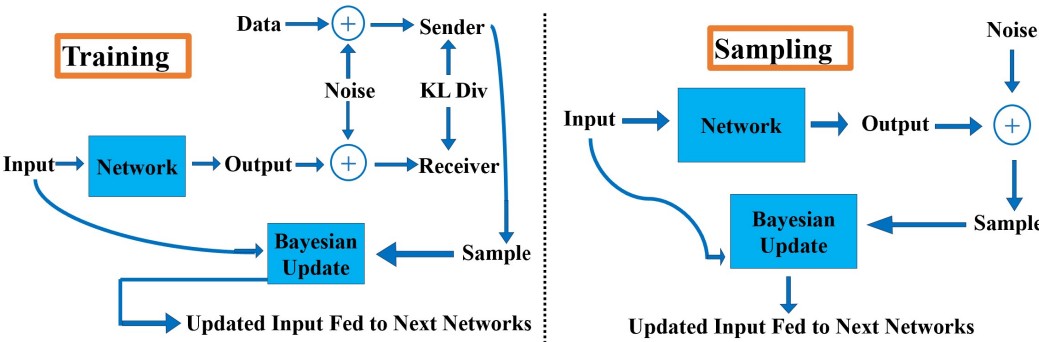

Figure 5: **Training (LHS):** Initially, the parameters of the input data distribution are input into the network, which subsequently yields the parameters of the output data distribution. An output sample is then drawn from this output distribution. To create the sender and receiver distributions, identical noise - as determined by the accuracy schedule - is applied to both the original data and the output sample. The Kullback-Leibler (KL) divergence between these two distributions is computed to formulate the loss. Following this, a sample from the sender distribution is utilized to revise the original input distribution via Bayesian update before re-entering the network. This updated distribution then serves as the new input for the subsequent network iteration. This cycle is repeated $N$ times, with the continuous time loss function emerging from extending the time parameter to infinity. **Sampling (RHS):** The sampling process mirrors training, with a notable distinction: noise is added solely to the output sample. This noise-modified sample is then used to update the initial input distribution through Bayesian updating.

---

**Algorithm 1** Continuous Output Predictioon

---

Note that $\boldsymbol{\theta} = \{\boldsymbol{\mu}, \rho\}$, but $\rho$ is fully determined by $t$
For our experiments $t_{min} = 1\mathrm{e}{-}10$, $[x_{min}, x_{max}] = [-1, 1]$
**Input:** $\boldsymbol{\mu} \in \mathbb{R}^D, t \in [0, 1], \gamma >\in \mathbb{R}^+, t_{min} \in \mathbb{R}^+, x_{min}, x_{max} \in \mathbb{R}$
**if** $t < t_{min}$ **then**
    $\hat{\mathbf{x}}(\boldsymbol{\theta}, t) \leftarrow 0$
**else**
    Input $(\boldsymbol{\mu}, t)$ to network, receive $\hat{\boldsymbol{\epsilon}}(\boldsymbol{\theta}, t)$ as output
    $\hat{\mathbf{x}}(\boldsymbol{\theta}, t) \leftarrow \frac{\mu}{\gamma} - \sqrt{\frac{1-\gamma}{\gamma}}\hat{\boldsymbol{\epsilon}}(\boldsymbol{\theta}, t)$
    clip $\hat{\mathbf{x}}(\boldsymbol{\theta}, t)$ to $[x_{min}, x_{max}]$
**end if**
**Return** $\hat{\mathbf{x}}(\boldsymbol{\theta}, t)$

---

**Algorithm 2** Continuous-Time-Loss for Guided-BFNs

---

**Input:** $\sigma_1 \in \mathbb{R}^+$, continuous data $\mathbf{x} \in \mathbb{R}^D$, conditions $h(\tau)$, scalar $\alpha = 1$
$t \sim U(0, 1)$
$\gamma \leftarrow 1 - \sigma_1^{2t}$
$\boldsymbol{\mu} \sim \mathcal{N}(\gamma\mathbf{x}, \gamma(1 - \gamma)\boldsymbol{I})$
$\boldsymbol{\mu} \leftarrow \alpha\boldsymbol{\mu}h(\tau)$ # conditional guidance
$\hat{\mathbf{x}}(\boldsymbol{\theta}, t) \leftarrow \text{CTS\_OUTPUT\_PREDICTION}(\boldsymbol{\mu}, t, \gamma)$ # conditional guidance
$\hat{\mathbf{x}}(\boldsymbol{\theta}, t) \leftarrow \alpha\hat{\mathbf{x}}(\boldsymbol{\theta}, t)h(\tau)$
$\mathbf{x} \leftarrow \alpha\mathbf{x}h(\tau)$
$L^\infty(\mathbf{x}) \leftarrow -\ln \sigma_1 \sigma_1^{-2t} \|\mathbf{x} - \hat{\mathbf{x}}(\boldsymbol{\theta}, t)\|^2$

---

---

**Algorithm 3** Trajectory Generation in Guided-BFNs

---

**Input:** $\sigma_1 \in \mathbb{R}^+$, number of steps $n \in \mathbb{N}$, conditions $h(\tau)$, scalar $\alpha, s$, reward model $\mathcal{J}_\phi$
$\boldsymbol{\mu}, \hat{\mathbf{x}}(\boldsymbol{\theta}, t)_{pre} \leftarrow \mathbf{0}$
$\rho \leftarrow 1$
\# Take $m(m < n)$ points at equal intervals for interpolation acceleration sampling
**for** $i = 1$ **to** $n$ **do**
   $t \leftarrow \frac{i-1}{n}$
   $\hat{\mathbf{x}}(\boldsymbol{\theta}, t) \leftarrow \text{CTS\_OUTPUT\_PREDICTION}(\boldsymbol{\mu}, t, 1 - \sigma_1^{2t})$
   **if** $i > 1$ **then**
      $\hat{\mathbf{x}}(\boldsymbol{\theta}, t) \leftarrow \hat{\mathbf{x}}(\boldsymbol{\theta}, t) + s\boldsymbol{\sigma_1^t} \nabla \mathcal{J}_\phi(\hat{\mathbf{x}}(\boldsymbol{\theta}, t)_{pre})$ \# gradient guidance
   **end if**
   $\hat{\mathbf{x}}(\boldsymbol{\theta}, t) \leftarrow \alpha \hat{\mathbf{x}}(\boldsymbol{\theta}, t) h(\tau)$ \# conditional guidance
   $\alpha \leftarrow \sigma_1^{-2i/n} \left( 1 - \sigma_1^{2/n} \right)$
   $\mathbf{y} \sim \mathcal{N}(\hat{\mathbf{x}}(\boldsymbol{\theta}, t), \alpha^{-1}\boldsymbol{I})$
   **if** $i > 0$ **then**
      $\boldsymbol{\mu} \leftarrow \boldsymbol{\mu} + s\boldsymbol{\sigma_1^t} \nabla \mathcal{J}_\phi(\hat{\mathbf{x}}(\boldsymbol{\theta}, t)_{pre})$ \# gradient guidance
      $\boldsymbol{y} \leftarrow \boldsymbol{y} + s\boldsymbol{\sigma_1^t} \nabla \mathcal{J}_\phi(\hat{\mathbf{x}}(\boldsymbol{\theta}, t)_{pre})$ \# gradient guidance
   **end if**
   $\boldsymbol{\mu} \leftarrow \alpha \boldsymbol{\mu} h(\tau)$ \# conditional guidance
   $\boldsymbol{y} \leftarrow \alpha \boldsymbol{y} h(\tau)$ \# conditional guidance
   $\boldsymbol{\mu} \leftarrow \frac{\rho \boldsymbol{\mu} + \alpha \mathbf{y}}{\rho + \alpha}$
   $\rho \leftarrow \rho + \alpha$
   $\hat{\mathbf{x}}(\boldsymbol{\theta}, t)_{pre} \leftarrow \hat{\mathbf{x}}(\boldsymbol{\theta}, t)$
**end for**
$\hat{\mathbf{x}}(\boldsymbol{\theta}, 1) \leftarrow \text{CTS\_OUTPUT\_PREDICTION}(\mu, 1, 1 - \sigma_1^2)$
$\hat{\mathbf{x}}(\boldsymbol{\theta}, 1) \leftarrow \hat{\mathbf{x}}(\boldsymbol{\theta}, 1) + s\boldsymbol{\sigma_1} \nabla \mathcal{J}_\phi(\hat{\mathbf{x}}(\boldsymbol{\theta}, t)_{pre})$ \# gradient guidance
$\hat{\mathbf{x}}(\boldsymbol{\theta}, 1) \leftarrow \alpha \hat{\mathbf{x}}(\boldsymbol{\theta}, 1) h(\tau)$ \# conditional guidance
**Return** $\hat{\mathbf{x}}(\boldsymbol{\theta}, 1)$

---

### B.2 PSEUDOCODE FOR GUIDED-BFNS

In Guided-BFNs, conditional guidance is added in both training (algorithms 1 and 2) and sampling (algorithms 1 and 3), while gradient guidance is added only in sampling process. Interpolation acceleration sampling is also shown in algorithm 3.

## C SETTINGS

### C.1 DATASETS

The Maze-2D dataset consists of three sub-datasets based on scale: Umaze, Medium, and Large, which require traversing to a goal location where a reward of 1 is given. No reward shaping is provided at any other location.

The Kuka dataset focuses on block stacking tasks, where trajectories of a robotic arm are generated to manipulate blocks. This dataset includes three sub-datasets: Unconditional Stacking, for which the task is to build a block tower as tall as possible; Conditional Stacking, for which the task is to construct a block tower with a specified order of blocks; Rearrangement, for which the task is to match a set of reference blocks' locations in a novel arrangement. We train Guided-BFNs on 10000 trajectories from demonstrations generated by PDDLStream Garrett et al. (2022)); rewards are equal to one upon successful stack placements and zero otherwise. We use one trained Guided-BFNs for all block-stacking tasks, only modifying the perturbation function $h(\boldsymbol{\tau})$ between settings. In the Unconditional Stacking task, we directly sample from the unperturbed output distribution $p_O(\mathbf{x} \mid \boldsymbol{\theta}; t)$ to emulate the PDDLStream controller. In the Conditional Stackingand Rearrangement tasks, we compose two perturbation functions $h(\boldsymbol{\tau})$ to bias the sampled trajectories: the first maximizes the likelihood of the trajectory's final state matching the goal configuration, and the second enforces a contact constraint between the end effector and a cube during stacking motions.

**Final State Matching**  To enforce a final state consisting of block A on top of block B, we trained a perturbation function $h_{\text{match}}(\boldsymbol{\tau})$ as a per-timestep classifier determining whether a a state $s$ exhibits a stack of block A on top of block B. We train the classifier on the demonstration data as the diffusion model.

**Contact Constraint**  To guide the Kuka arm to stack block A on top of block B, we construct a perturbation function $h_{\text{contact}}(\boldsymbol{\tau}) = \sum_{i=0}^{64} -1 * \|\boldsymbol{\tau}_{c_i} - 1\|^2$, where $\boldsymbol{\tau}_{c_i}$ corresponds to the underlying dimension in state $\boldsymbol{\tau}_{s_i}$ that specifies the presence or absence of contact between the Kuka arm and block A. We apply the contact constraint between the Kuka arm and block A for the first 64 timesteps in a trajectory, corresponding to initial contact with block A in a plan.

The Locomotion dataset, an offline RL dataset, comprises nine sub-datasets containing trajectories related to various movement scenarios. We guide the trajectories generated by Guided-BFNs toward high-reward regions using the sampling procedure and condition the trajectories on the current state using the inpainting procedure. The reward function $J_\phi$ is trained on the same trajectories as Guided-BFNs.

### C.2 MODEL ARCHITECTURE

The neural network model architecture for Guided-BFNs is illustrated in fig. 6. The Guided-BFNs architecture comprises a U-Net structure with 6 repeated residual blocks. Each block consists of two temporal convolutions, each followed by group norm Wu & He (2018), and a final Mish nonlinearity Misra (2019). Timestep embeddings are generated by a single fully-connected layer and added to the activations of the first temporal convolution within each block.

### C.3 TRAINING SETTING

The model is trained with batch size of 32. The optimizer used is AdamW Loshchilov & Hutter (2017) with a learning rate of 0.0001, weight decay of 0.01, and $(\beta 1, \beta 2) = (0.9, 0.98)$.

The training process involves 500k steps. The reward function $\mathcal{J}_\phi$ mirrors the structure of the first half of the U-Net used for Guided-BFNs, concluding with a final linear layer that produces a scalar

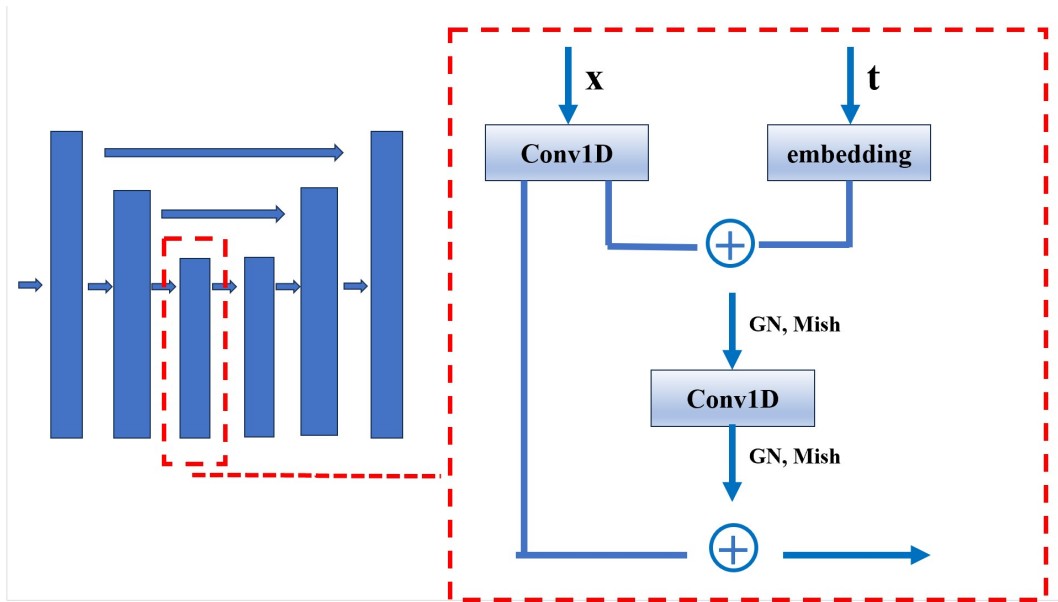

Figure 6: **TemporalUNet**: a model consisting of repeated (temporal) convolutional residual blocks. The overall architecture resembles the types of U-Nets with two-dimensional spatial convolutions replaced by one-dimensional temporal convolutions.

output. A planning horizon $T$ of 32 is employed in all locomotion tasks, 128 for block-stacking, 128 in `Maze2d-umaze`, 256 in `Maze2d-medium`, and 384 in `Maze2d-large`.

## D    OTHER EXPERIMENTS

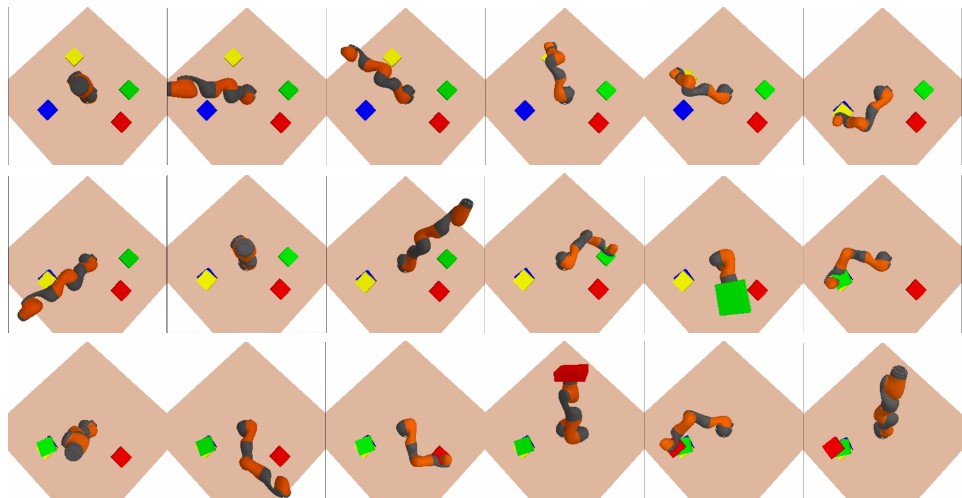

Table 3: Here is a demonstration showcasing the functionality of Guided-BFNs on the Kuka dataset. In the series of images from the top left to the bottom right, the robotic arm progressively attaches each block and positions it above the others, guided by the trajectories generated by Guided-BFNs. In the unconditional scenario, the objective is to stack the blocks to achieve the maximum height. On the other hand, the Kuka-conditional task involves additional conditions, such as the color order of the blocks, necessitating the use of gradient guidance to accomplish the desired stacking arrangement.

Table 4: The following table illustrates the increase in planning score with the escalation of planning time steps ($t_p$) in offline RL environments, specifically in the D4RL setting.

| Locomotion Datasets | Planning Score (Time Steps) | | | | | | | | | | |
|---|---|---|---|---|---|---|---|---|---|---|---|
| | 0 | 999 | 1999 | 2999 | 3999 | 4999 | 5999 | 6999 | 7999 | 8999 | 9999 |
| halfcheetah-medium-expert-v2 | 0.0225 | 0.3968 | 0.7849 | 1.1764 | 1.5713 | 1.8685 | 1.8985 | 2.2821 | 2.5318 | 2.9318 | 3.3389 |
| halfcheetah-medium-replay-v2 | 0.0226 | 0.2536 | 0.5071 | 0.759 | 1.0 | 1.2451 | 1.4888 | 1.7288 | 1.9823 | 2.1992 | 2.4315 |
| halfcheetah-medium-v2 | 0.0225 | 0.3312 | 0.641 | 0.969 | 1.3019 | 1.626 | 1.9414 | 2.2712 | 2.4634 | 2.4573 | 2.4496 |
| hopper-medium-expert-v2 | 0.0065 | 0.4927 | 0.7996 | 1.1065 | 1.4135 | 1.7204 | 2.0274 | 2.3343 | 2.6411 | 2.9481 | 3.2551 |
| hopper-medium-replay-v2 | 0.0065 | 0.3886 | 0.6942 | 0.9998 | 1.3055 | 1.6114 | 1.9166 | 2.2227 | 2.5276 | 2.8331 | 3.1382 |
| hopper-medium-v2 | 0.0065 | 0.5468 | 0.8537 | 1.1606 | 1.4676 | 1.7745 | 2.0815 | 2.3884 | 2.6953 | 3.0023 | 3.3092 |
| walker2d-medium-expert-v2 | -0.0001 | 0.314 | 0.5319 | 0.7498 | 0.9675 | 1.1855 | 1.4037 | 1.6224 | 1.8402 | 2.0581 | 2.2766 |
| walker2d-medium-replay-v2 | -0.0001 | 0.3073 | 0.5241 | 0.7414 | 0.9586 | 1.176 | 1.3931 | 1.6102 | 1.8273 | 2.0447 | 2.2617 |
| walker2d-medium-v2 | -0.0001 | 0.2925 | 0.5092 | 0.7263 | 0.9432 | 1.1601 | 1.377 | 1.5944 | 1.8113 | 2.0287 | 2.2461 |

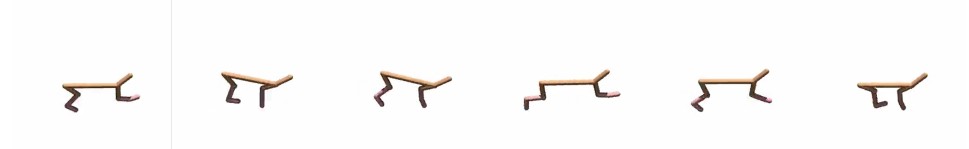

Table 5: This demonstration showcases the trajectory generated by Guided-BFNs to control the movements of the agent in the HalfCheetah environment of the D4RL offline RL dataset. The sequential movements depicted in the trajectory illustrate the effectiveness of Guided-BFNs in shaping the agent's behavior in navigating and interacting within the challenging dynamics of the HalfCheetah environment.

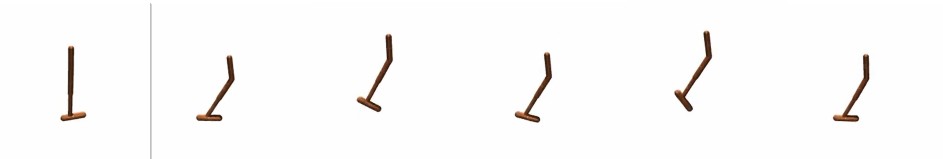

Table 6: This demonstration showcases the trajectory generated by Guided-BFNs to control the movements of the agent in the Hopper environment of the D4RL offline RL dataset. The sequential movements depicted in the trajectory illustrate the effectiveness of Guided-BFNs in shaping the agent's behavior in navigating and interacting within the challenging dynamics of the Hopper environment.

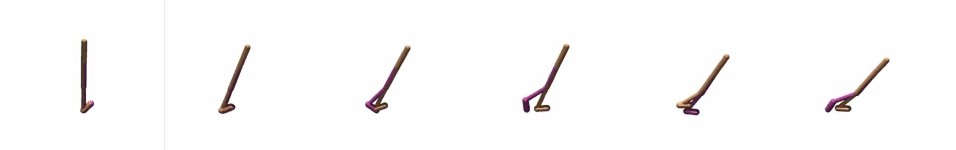

Table 7: This demonstration showcases the trajectory generated by Guided-BFNs to control the movements of the agent in the Walker2d environment of the D4RL offline RL dataset. The sequential movements depicted in the trajectory illustrate the effectiveness of Guided-BFNs in shaping the agent's behavior in navigating and interacting within the challenging dynamics of the Walker2d environment.

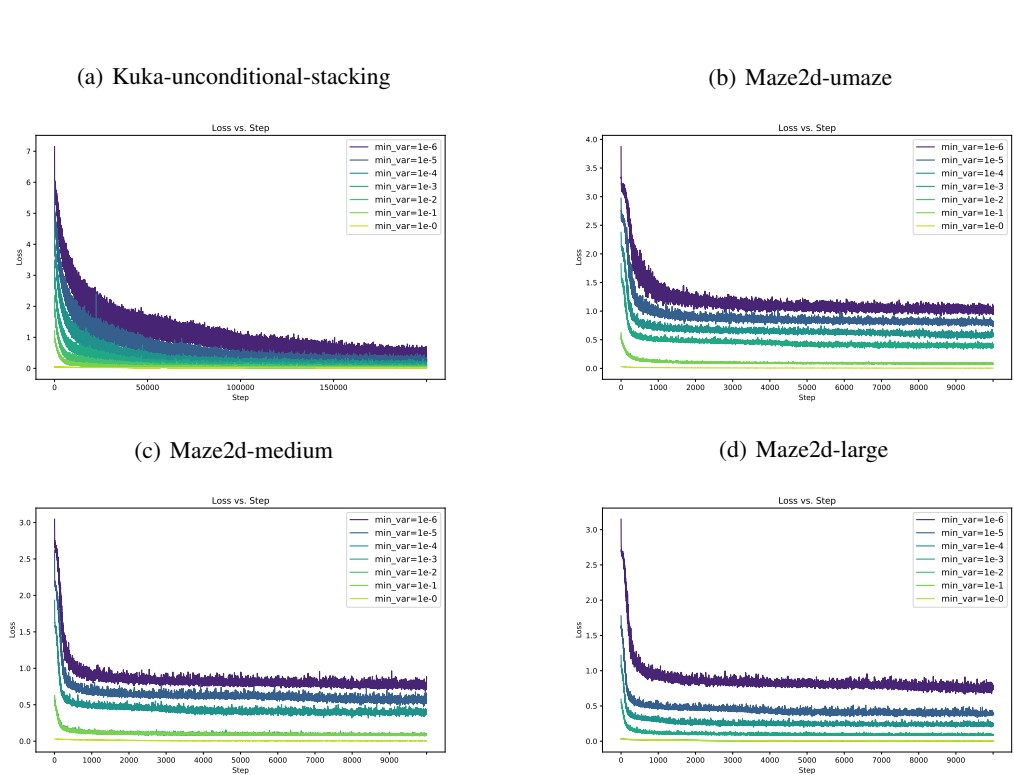

(a) Kuka-unconditional-stacking       (b) Maze2d-umaze

(c) Maze2d-medium          (d) Maze2d-large

Table 8: Loss across different $\sigma_1$ values in various datasets throughout training. In our observations, we note that the converged training loss tends to decrease as $\sigma_1$ increases. Simultaneously, the fluctuation of loss during training shows an inverse relationship with $\sigma_1$; it increases as $\sigma_1$ decreases.

$\sigma_1 = 0.9$  $\sigma_1 = 0.1$  $\sigma_1 = 0.01$  $\sigma_1 = 0.001$  $\sigma_1 = 0.0001$ $\sigma_1 = 0.00001$ $\sigma_1 = 0.000001$

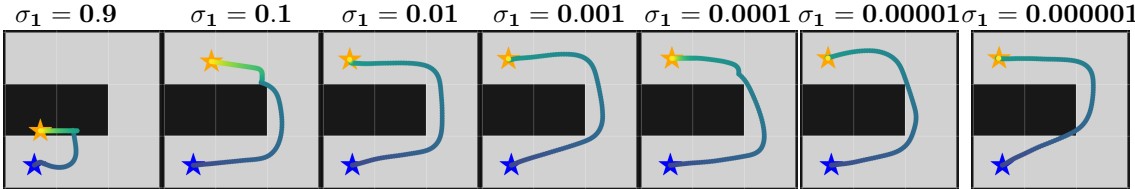

Table 9: Visualizations on Maze2D-Umaze are employed to intuitively demonstrate how $\sigma_1$ influences planning performance by generating trajectories.

