# OpenReview forum: "Guided-BFNs: Towards Visualizing and Understanding Bayesian Flow Networks in the Context of Trajectory Planning"
_ICLR.cc/2025/Conference — Submitted to ICLR 2025_

### Official Review · Reviewer_Hzsv · 2024-11-03

**Soundness:** 2
**Presentation:** 2
**Contribution:** 2
**Rating:** 5
**Confidence:** 3

**Summary:**

This paper extends Bayesian Flow Networks (BFNs) to trajectory planning, introducing a new approach called "Guided-BFNs," where "guided" refers to conditional and gradient-based guidance. Through ablation studies, the paper identifies key parameters and components of the BFN structure. Both qualitative and quantitative comparisons demonstrate that the proposed Guided-BFNs outperform the baseline diffuser [Janner et al. 2022] in most scenarios.

**Strengths:**

- The first to apply BFN in trajectory planning setting
- It demonstrates competitive performance in experiments

**Weaknesses:**

The method seems largely based on combining BFN [Graves et al., 2023] with techniques from the Diffuser [Janner et al., 2022]. Beyond the interpolation-based sampling method, I'm uncertain if there are any novel components introduced in this work.

Despite the authors' efforts to compare their method with the baseline Diffuser, there is limited  evidence suggesting clear scenarios where one method would be preferred over the other. Although quantitative results are presented in Table 1, the number of runs, or the variance never seem to have been mentioned, making it hard to assess the statistical significance of these differences.Further details and insights would strengthen the paper. For instance, Fig. 3 (Left) indicates that BFN learns faster than DM and produces smoother trajectories. What might contribute to this difference? Additionally, while the paper asserts that BFN lacks the sampling step restrictions present in DM, the performance shown in Fig. 2(a) does not appear to improve monotonically as sample steps increase. How would the performance scale with more than 256 sample steps?

Other areas for improvement:

- Network architecture is the same as the one in diffuser but lacks citation.
- Fig 2 does not specify which optimization step the checkpoints were taken from.
- Fig 2 lacks planning performance metrics for different training checkpoints and the num of planning steps. Visually it’s  difficult to determine if  performance  increase from planning time steps 150, 200, and 250.
- Fig 2 caption states: “This finding enlightens similar results in the original testbed of image and text data in BFNs” but does not cite supporting evidence.

Typos:

- line 72, capital letter “We”
- line 94, “a” interpolation
- Figure 4 left, x label should be alpha’ instead of alpha

**Questions:**

- Regarding the computational complexity, why is it that the Guided-BFNs is only a little slower than diffuser when it has more than 7 times FLOPs?
- In Fig. 2(c), what does the “i” in legend refer to?

---

### Official Review · Reviewer_UQyj · 2024-11-03

**Soundness:** 2
**Presentation:** 2
**Contribution:** 3
**Rating:** 5
**Confidence:** 4

**Summary:**

The authors present work that extends Bayesian Flow Networks with two functionalities: conditioning and guidance through a gradient function. Two elements that are required to apply the technique to a diffusion like planner such as "Diffuser" introduced by Janner et al. The authors suggest mechanisms for these methods and then continue to evaluate their algorithm on various RL tasks, executed through trajectory sampling. The authors report competitive metrics on various RL benchmarks.

**Strengths:**

I think this paper has the right ingredients to be a good paper
- The paper builds on interesting previous work
- It extends this previous work with functionality (conditioning, guidance) that enables its application to a useful problem (RL in the "guided conditional sampling" style)
- The paper reports competitive performance

**Weaknesses:**

- BFN's is not a widely adopted mechanism, published in 2023, it has only 22 citations at the time of reviewing. I would hope the authors spend some more time in explaining the (quite complex) mechanism of BFN's and why they are expected to be better than diffusion models (since they are ultimately similar), and why the experimental results show such a large difference. There is a mathematical overview in the appendix, but I would hope to see some more in the main text too. In particular I think the authors should focus on explaining why their additions/modifications to the algorithm are the right choice.
- The discussion in the appendix matches so closely to the original text and math of Graves et al that it would be good to cite the original work a few extra times.
Some of the presentation in the paper is quite messy, in particular in relation to the math describing the mechanism the authors develop. They could potentially be incorrect:
- Equation 7 seems like perhaps the authors mean $p_{O} \leftarrow p_{O}\ h(\tau)$ ? It is not clear what the proportionality in that equation means. It seems odd to have an identical $p_O$ left and right. Perhaps it is a typo?
- The unlabelled equation (between 8 and 9 on page 4) is not a correct specification of a delta function. This should be addressed.
- Generally, it appears as if between equation 8 and 9, a Dirac delta is applied. However, equation 4 shows the distribution is already a Dirac delta. This leads to some kind of undefined output. I.e. there is no mass on any other value than $\hat{x}$, so replacing the values as the authors do is inconsistent with the BFN method.
- In section 4.2, the unlabelled equation seems to suggest that $\alpha = h(\tau)$, but this is not what it says in the text. From the text it seems to appear as if they are interpolating between the sampled and the constrained value with a factor $\alpha$, but this is not what the unlabelled equation says. The authors should address this and clarify.

**Questions:**

- It is not completely clear to me the logical place to insert gradient information is in the sender distribution. The authors do make an argument about this, but it seems to me as if this should be something on the receiver side? Can the authors comment? I think some more elaboration on that design choice would improve the paper.
- There is no axis label in Fig 3 left. Is this as a function of training time? Or something else?
- The paper by "Graves, 2023" et al states: "Because the focus of this paper is on probabilistic modelling rather than image generation, FID scores were not calculated. However, examples of generated data are provided for all experiments.". This seems to suggest that we can not expect competitive metrics from the method, since it does not target that. The results presented in this paper seem to vastly improve over diffusion models. Why is that the case?
- In the appendix, in table 2, it appears as if training the network takes only ~17 seconds, and for diffuser it takes ~16s. Is that correct, or am I missunderstanding this? If this is not the case, can the authors give a measure of the amount of compute used (flops, wall clock time, .. etc) that is required to train the network to convergence?

---

### Official Review · Reviewer_9exq · 2024-11-04

**Soundness:** 2
**Presentation:** 2
**Contribution:** 2
**Rating:** 3
**Confidence:** 3

**Summary:**

This paper proposes Guided-BFNs, which combine Bayesian Flow Networks (BFNs) and training-free conditional guidance for model sampling and trajectory plannings. This paper conduct a Comparative analysis of trajectories sampled from DMs and BFNs, and ablation studies to investigate how to conditional scaling factors and gradient scaling factors affect the trajectory planing performance.

**Strengths:**

1. The background to introduce the BFNs is clear.
2. Figures and diagrams for visualization and demonstrating proposed approach are clear.

**Weaknesses:**

1. From my point of view, the Bayesian Flow Network(BFN) is structured as a more general framework for the diffusion model by following appendix, as it has similar loss function given by Equation (3) and ideas of adding noise. Section 3 ``Guided-BFNs'' describes a training-free guidance, which is fairly close to what has been proposed in [1], in particular, the definition of $h(\tau_t\})$ in section 3.1, and the guidance $g$ in section 3.2, are almost identical to the concept in [1]. In conclusion, this paper proposes to combine the BFNs and the guidance from [1], and the novelty is not quite apparent to me in this case.
2. Following the above point, this paper misses some contents in background or the related work to introduce the existing training-free guidance approaches.
3. In Figure 3, it's been commented that the BFNs have more noisy initial data than DMs, which appears a bit strange to me. DMs are supposed to have quite noisy initial data at the beginning, e.g. Figure 4 from [1]. I'm wondering if there's any misunderstanding or potential misimplementation here. Correct me if I am wrong, if the last column in the left figure represents generated samples at the final step, I find the trajectory produced by DMs more appealing. The trajectory looks more natural because it does not get excessively close to the obstacles' boundaries, unlike the one generated by BNFs.


[1] Janner, Michael, et al. "Planning with Diffusion for Flexible Behavior Synthesis." International Conference on Machine Learning. PMLR, 2022.

**Questions:**

1. In 4.3, ``In contrast, DMs impose limitations, permitting only a finite number of sampling steps equal to those fixed during the training process.'' I do not agree with this. In general, during the training, noisy level can be randomly sampled from the diffusion schedule and it does not have to fixed, therefore, the number of sampling steps is not fixed by the training. Kindly correct me I miss understand the idea here.
2. In Figure 3, the second row on the right, why would BFNs generate the sample that the yellow star is right in the obstacle when $\alpha\to 0$? Is this simply some initial default state?
3. This could be minor: from the notation side, why is sender distribution denoted as $P_S(\cdot | \mathbf{x}; \alpha\mathbf{I})$ and the receiver distribution denoted as $P_R(\mathbf{y} | \boldsymbol{\theta}; t, \alpha)$ on line 153? I just found it inconsistent.

---

### Official Review · Reviewer_g4wA · 2024-11-04

**Soundness:** 2
**Presentation:** 2
**Contribution:** 1
**Rating:** 3
**Confidence:** 5

**Summary:**

This paper proposes Guided-BFNs, which combine BFNs with conditional guidance and gradient guidance to form a guided planning algorithm. Then, the authors compare Guided-BFN with Diffuser and conducts experiments on D4RL's Mujoco, Maze2d, and Kuka environments, surpassing the baseline algorithm Diffuser.

**Strengths:**

1. The authors have introduced a new generative model for decision-making tasks, which is very interesting. The reviewers are eagerly looking forward to seeing new generative models being applied to decision-making problems, as this is an open and meaningful issue.

2. Overall, the paper is well writen, and the reviewer can understand how BFN is integrated with the planning algorithm

3. The experiments show that Guided-BFNs can surpass the baseline algo. Diffuser in many d4rl environments

**Weaknesses:**

1. Although the authors presented several analyses demonstrating the innovation and advantages of BFN, such as conditional guidance and gradient guidance, the reviewers believe that the core advantages of BFN compared to diffusion models are still uncertain. Conditional guidance and gradient guidance can be seen as a combination of classifier-free and classifier-based diffusion models. Is BFN a direct replacement for diffusion models? What significant advantages does it have?
2. Based on the first point, this paper also lacks experimental comparisons. The reviewers think that the core of this paper is a new backbone algorithm—BFN—which replaces the diffusion model (other parts of the planning algorithm are very similar to Diffuser). However, the paper neglects to compare with other generative models or architectures, such as Decision Diffuser, Decision Transformer, GAN, MAMBA or other offline rl baseline.
3. Given that Diffuser is a pioneering diffusion model framework and has achieved many improvements, can BFN be combined with some of them to enhance performance?
4. Lack of related work. If BFN is an improvement based on diffusion models, then more diffusion model works should be discussed. If BFN is a fundamental architecture, then more foundational architecture works should be introduced.
5. BFN has too many hyperparameters that need to be tuned, posing challenges to the method's usability.

**Questions:**

Some experimental settings are unclear, such as why the experimental results do not have mean and variance. Were the experiments averaged over several seeds?
In the visualization of Figure 3, BFN and Diffuser exhibit the same performance. Please explain the advantages of BFN in this figure.

---

### Meta-Review · Area_Chair_7uAh · 2024-12-17

**Metareview:**

This paper presents a method for facilitating the application of Bayesian Flow Networks through the use of conditional and gradient guidance. The paper has received several critical comments from Reviewers about the relevance of the analysis, discussion of related works, and overall impact of this work. In particular, the empirical analysis has been significantly criticised by most Reviewers.

The initial negative assessment of Reviewers have not been changed during the discussion, especially considering the absence of a rebuttal from the Authors.

I recommend the Authors to address Reviewers' concerns for future submissions.

**Additional Comments On Reviewer Discussion:**

No discussion has been needed, given the unanimous assessment for this paper and the absence of Authors' rebuttal.

---

### Decision · Program_Chairs · 2025-01-22

Reject